# Design of a Full-Ocean-Depth Macroorganism Pressure-Retaining Sampler and Fluid Simulation of the Sampling Process

Guangping Liu, Yongping Jin *, Youduo Peng, Deshun Liu and Buyan Wan

National-Local Joint Engineering Laboratory of Marine Mineral Resources Exploration Equipment and Safety Technology, Hunan University of Science and Technology, Xiangtan 411201, China
* Correspondence: jinyongping@hnust.edu.cn; Tel.: +86-155-8019-2972

**Abstract:** Hadal seafloor organisms live under ultra-high pressure, in low temperatures, and other environments for a long time, which puts higher requirements on the structural design of deep-sea biological samplers. In this paper, we present a full-ocean-depth hydraulic suction macroorganism pressure-retaining sampling method, which actively samples seafloor macroorganisms by pumping and stabilizing the pressure inside the sampler using a pressure compensator. Firstly, the structure and working principle of the hydraulic suction macroorganism pressure-retaining sampler (HSMPS) were introduced. Then the flow field of the HSMPS sampling process was analyzed, and the velocity and pressure distribution of the flow field at different locations of the HSMPS were obtained. In response to the problem of the low viability of samples collected by deep-sea biological samplers, the changes in radial velocity and pressure at different positions of the sampler under different pumping flows were analyzed. Finally, the appropriate suction flow rate was selected based on the analysis results, and HSMPS suction tests and simulated sampling tests, under a 110 MPa high-pressure environment, were carried out using the developed HSMPS engineering prototype. The test results verify the feasibility of the HSMPS design, which will provide strong support for the deep abyssal seafloor sampling operation of the full-ocean-depth manned submersible.

**Keywords:** deep sea environment; pressure-retaining sampling; structure design; fluid simulation

## 1. Introduction

The seafloor at full ocean depth (about 11,000 m) has a wide variety of macroorganism resources and new biological species, and obtaining live macroorganisms on the seafloor at full ocean depth is a prerequisite for scientific research on environmental changes, the evolutionary processes of life, the distribution of macroorganism species, and the survival of macroorganisms on the seafloor at full ocean depth [1–3]. Deep-sea biological sampling technology can currently be achieved using passive trap samplers and active pumping samplers. When operating passive trap samplers, the sampler is generally mounted on the lander and samples deep-sea organisms using bait trapping [4]. After the sampling is completed, the lander and sampler are recovered to the surface together by the command on the deck. The sampler is highly dependent on the mother ship and susceptible to ocean currents, which cannot precisely locate the sampling point, and the samples collected are random, and there is a possibility that the samples cannot be collected in one dive. When operating a pumped sampler, the sampler can be carried on a manned/unmanned submersible and uses a suction pump to generate negative pressure for the active sampling of seafloor organisms [5]. A precise sampling of seafloor organisms is achieved using a manipulator on the submersible. The sampler can precisely locate sampling points and provide timely feedback, enabling the sampling of different species of organisms in the deep sea with high sampling rates.

Deep-sea biological sampler equipment is a necessary technical means to research the seafloor environment and carry out sampling and exploration of the seafloor's biological resources. Since the 20th century, many biological samplers have been developed at home and abroad. For the earliest sampling systems, please refer to Brown et al. [6], Yayanos et al. [7], Phleger et al. [8], and Wilson et al. [9]. For a historical review of biological samples, please refer to Feng et al. [10]. Recently, Shilito [11] developed a PERISCOP system for an active sampling of deep-sea organisms using a suction device, followed by preservation using a pressure-retaining recovery device, which successfully obtained a live fish at a water depth of 2300 m. Billings et al. [12] developed a SyPRID sampler, and the SyPRID sampler can be independently controlled to perform two repetitions or two independent samples each time the AUV Sentry performs its task. Peoples [13] developed a new lander system that is capable of sampling in a full-ocean-depth environment. The lander system has a modular design and instruments for collecting macrofauna and flora, water, and taking videos and images, and a pressure-retaining deep-sea macroorganism sampler can be deployed on the lander. Wang [14] proposed a motor-driven piston pressure-retaining sampler, in which the sampling device traps seafloor macroorganisms onto a piston using bait, and then the piston is controlled by a motor to enter the deep-sea macroorganism sampler. Liu [15] designed a full-ocean-depth sediment gas-tight sampler, which mainly samples seafloor microorganisms and uses a pressure compensator to compensate for the pressure drop in the sampler recovery process.

Deep-sea organisms live in an environment with high hydrostatic pressure, low temperatures, high concentrations of inorganic matter, and low organic carbon content on the seafloor for a long time, and most deep-sea organisms have characteristics such as pressure and being thermophilic [16] . To obtain in situ seafloor biological samples, the deep-sea biological sampler needs to be designed for pressure retention and thermal insulation. In addition, considering that the sampler itself has no navigational capability, it needs to be carried on a deep submersible to dive to the specified sea depth when working, and the space and weight of the deep submersible for one dive are very valuable, so the appearance, size, weight, and other parameters of the sampler have severe requirements. As for the use environment of the deep-sea sampler, the sampler needs to work in a harsh environment of low temperatures and ultra-high-pressure force. When diving, the sampler is subjected to the external pressure of external seawater, and the sampler needs to withstand the ultra-high-pressure force of the seafloor, which reaches 110 MPa in the full-sea -depth environment, thus putting higher requirements on the structure and materials of the sampler. During recovery, the ambient pressure outside the sampler decreases, at which time the internal pressure of the sampler is greater than the external ambient pressure, and the cylinder expands and deforms under the pressure difference to produce a pressure drop, so a pressure compensation mechanism needs to be designed [17]. At present, most of the existing sampling equipment for deep-sea organisms samples deep-sea microorganisms. When sampling deep-sea macroorganisms, the rated depth of sampling is shallow, and passive trapping is adopted to sample deep-sea organisms, so it is impossible to accurately sample deep-sea organisms. Here, we propose a full-ocean-depth hydraulic suction macro-biological sampling method, which actively samples benthic organisms by generating negative pressure through suction pumps. The HSMPS can be carried on a manned/unmanned submersible and can sample macroorganisms at 11,000 m on the seafloor. The pressure compensator is used to compensate for the pressure drop in the recovery process of the HSMPS to realize the pressure-retaining sampling of deep-sea macroorganisms. The HSMPS can be sealed using a single trigger with a manipulator, which is easy to operate and reliable to seal.

The article is structured as follows. In Section 2, the structure and working principle of the HSMPS is described. In Section 3, the simulation method in the process of seafloor biology collection using the HSMPS is described, and the flow field pressure and velocity distributions at different locations of the HSMPS are obtained. In Section 4, the radial pressure and velocity at different positions of the HSMPS at different pumping speeds were

analyzed to obtain the flow field distribution law at each position and select reasonable pumping parameters. The HSMPS suction test and simulated sampling test were conducted under a 110 MPa high-pressure environment.

## 2. Structure and Working Principle of the HSMPS

### 2.1. Structure of the HSMPS

The HSMPS can be carried on HOVs and ROVs to capture seafloor macroorganism samples using remote or automated operations. The status of HSMPS operating with the submersible onboard is shown in Figure 1. HSMPS is mainly composed of three parts: pumping suction area, pressure-retaining area, and diversion area. The pumping suction area is mainly composed of a suction pump and hydraulic interface, through the work of the suction pump to generate negative pressure in the pressure-retaining area and diversion area to achieve sampling of deep-sea microorganisms. The suction pump is a hydraulically driven centrifugal pump (model 9303P-HM4C) manufactured by HYPRO, which is suitable for deep-sea environments and has a maximum pump flow of 18.2 m$^3$/h. The hydraulic source of the suction pump is provided by the submersible. The diversion area consists of a suction tube and a handle, and the capture of deep-sea organisms in any direction can be achieved using a robot grasping the handle on the suction tube. Suction pipe selection is PVC steel wire hose, with built-in spiral steel wire resistance to negative pressure and bending, expansion and contraction, good performance of heat and cold resistance, anti-aging, and other advantages. The inner diameter of the suction tube is 60 mm, which can achieve most macroorganism sampling in hadal environments. The HSMPS is mounted on a support frame, with the length, width, and height of the support frame being 700 × 300 × 160 mm. HSMPS has the following features: (1) HSMPS can control the suction rate during the capture of seafloor organisms; (2) it can maintain the in situ pressure of the sample; (3) all key components can withstand a pressure of 110 MPa; (4) it can be sealed in one trigger without external power and is easy to operate; and (5) it can achieve pressure drop-free transfer in the laboratory.

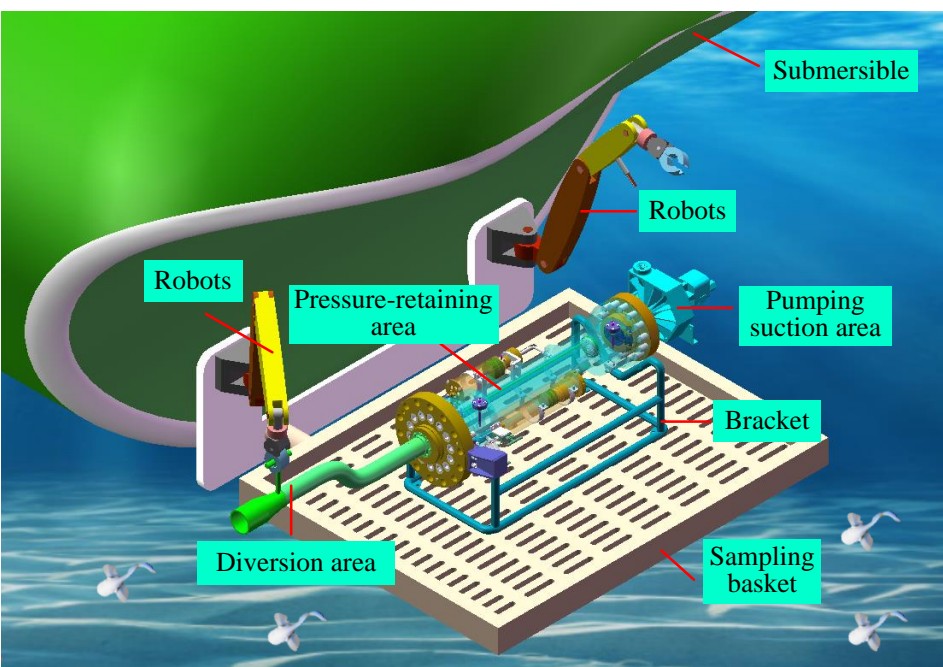

**Figure 1.** HSMPS onboard the submersible.

The structure of the HSMPS pressure-retaining area is shown in Figure 2, which mainly consists of outlet and inlet sealing valves, a pressure-retaining cylinder, a pressure compensator, a bait cartridge, a trigger lever, a transfer mechanism, a non-return device,

and several high-pressure valves. The inlet sealing valve is connected to the suction pipe in the diversion area through the interface, and the outlet sealing valve is connected to the suction pump in the pumping suction area through the interface; the outlet and inlet sealing valves are composed of the valve cover and valve body, which are sealed from inside to outside, and the valve cover and valve body are designed with eccentric structure, and the eccentric angle is 10°. Sealing of the HSMPS is achieved by pulling the trigger lever on the submersible to close the outlet and inlet sealing valves. The pressure-retaining cylinder material is TC4 titanium alloy, which is characterized by high strength, lightweight, and corrosion resistance. The role of the pressure-retaining cylinder is to provide a high-pressure environment for the collected biological samples, and the pressure-retaining cylinder can withstand a high pressure of 110 MPa. During HSMPS collection, seawater is discharged through a small hole in the non-return device, and the biological sample is trapped in the pressure-retaining cylinder. The pressure compensator consists of an end cap, piston, and barrel. The pressure compensator is pre-charged with a certain amount of nitrogen through a filling valve to compensate for the pressure drop during the HSMPS recovery to the deck. The bait cartridge is mainly composed of a cartridge, piston, one-way valve, etc. The food is placed on the piston, and through the extrusion of the piston, the food flows into the pressure-retaining cylinder through the one-way valve to provide food and nutrition for the collected macrobiotic samples. The transfer mechanism includes a handle, gear lever, and gear. After the HSMPS sampling is completed, the inlet sealing valve is used to dock with the biological culture device, and turning the handle drives the checker to move on the gear lever to realize the transfer of the biological sample inside the pressure-retaining cylinder. The structural parameters of HSMPS are shown in Table 1.

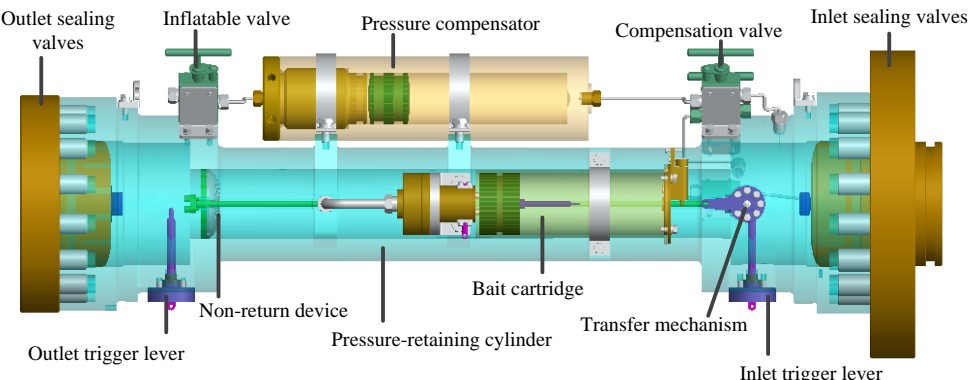

**Figure 2.** Structure diagram of pressure-retaining area.

**Table 1.** Structural parameters of HSMPS.

| Structures | Values | Structures | Values |
|---|---|---|---|
| Inner diameter of inlet sealing valve/$D_1$ | 60 mm | Length of pressure compensator/$L_2$ | 300 mm |
| Maximum internal diameter of sealing valve/$D_2$ | 68 mm | Inner diameter of pressure compensator/$D_6$ | 50 mm |
| Inner diameter of outlet seal valve/$D_3$ | 62 mm | Inner diameter of bait cartridge/$D_7$ | 60 mm |
| Length of pressure-retaining cylinder/$L_1$ | 526 mm | Length of bait cartridge/$L_3$ | 187 mm |
| Inner diameter of suction pipe/$D_4$ | 60 mm | Eccentric angle of sealed valve/$\theta$ | 10° |
| Inner diameter of pressure-retaining cylinder/$D_5$ | 68 mm | Maximum pumping flow rate/$Q$ | 18 m³/h |

### 2.2. Working Principle of HSMPS

The working principle of HSMPS is shown in Figure 3. The deep-sea macroorganism sampler captures deep-sea macroorganisms in three processes: lowering, sampling, and recycling.

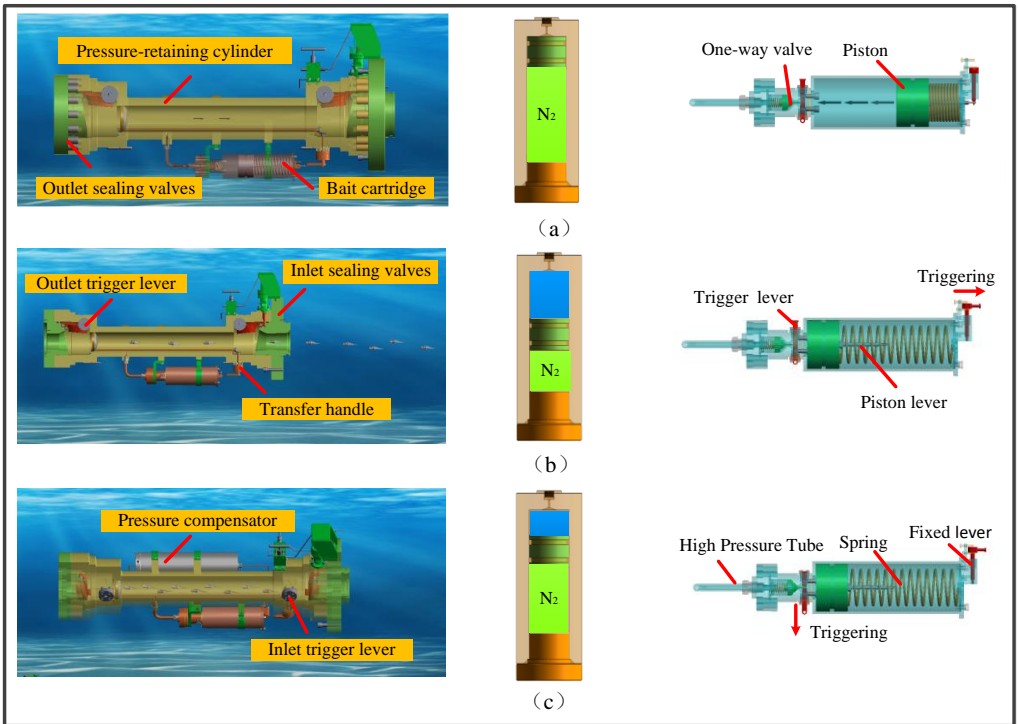

**Figure 3.** Working principle of HSMPS. (**a**) Lowering; (**b**) Sampling ; (**c**) Recycling.

(a) Lowering: The HSMPS components are installed on the submersible after integration. Before lowering the submersible, open the outlet and inlet sealing valves, and limit them through the trigger lever. Then, a certain amount of nitrogen is pre-charged into the pressure compensator through the filling valve so that the piston is at the top of the pressure compensator. Put the bait package into the top of the bait barrel piston; the top rod at the bottom of the check valve on the bait barrel is in contact with the level of the trigger rod, so that the check valve on the bait barrel is in the open state, the fixed rod is limiting the piston rod, and the spring is in the compressed state. The suction pump is connected to the hydraulic source on the submersible through a hydraulic line (Figure 3a).

(b) Sampling: During the dive of the submersible, the piston in the pressure compensator moves downward under the pressure of seawater until the pressure in the lower and upper chambers of the piston reach equilibrium. When the sampling point is reached, the deep-sea organisms are captured by the robot grabbing the handle on the suction pipe, triggering the hydraulic source button on the submersible to make the suction pump work, and the suction pump generates negative pressure to make the fish–water mixture enter the pressure-retaining cylinder through the suction pipe, and the seawater flows out from the suction pump outlet through the non-return device, and the macroorganisms are trapped in the pressure-retaining cylinder by the non-return device. The bait cartridge retaining lever is triggered by the robot to remove the restriction on the piston rod, causing the spring to drive the piston to compress the bait packet (Figure 3b).

(c) Recycling: HSMPS sampling is completed, and the inlet and outlet trigger lever is pulled by a robot to cancel the restriction on the outlet and inlet sealing valves to achieve the pressure-retaining cylinder seal. During the recovery of HSMPS to the

deck, the pressure-retaining cylinder expands and deforms due to the reduction in external seawater pressure, at which time the pressure compensator will compensate for the pressure loss caused by the expansion and deformation of the pressure-retaining cylinder, and the piston moves upward. The piston in the bait cylinder is driven by a spring to compress the bait packet, and the bait flows from the check valve into the pressure-retaining cylinder to provide nutrients to the macroorganisms (Figure 3c).

## 3. Description of the Simulation of Method

### 3.1. Modeling Details

The flow field of the HSMPS capture process was simulated numerically, and the HSMPS model was simplified to retain only the pressure-retaining and inflow regions, and the simplified computational model is shown in Figure 4. The flow field is divided into two calculation domains: the internal flow field and the external flow field. The internal flow field includes the flow field of the pressure-retaining area and the diversion area. The external flow field is all the areas except the internal flow field, and the external flow field size is 1000 × 2000 × 2000 mm (length × width × height). The left end of the outflow field coincides with the plane of the exit end of the HSMPS, and the center of the circle at the exit end is defined as the cartesian coordinate origin, and the distance between the hadal snailfish and the inlet of the suction tube in the diversion area is 1 mm. The inlet boundary condition is defined as the mass-flow outlet, the outlet boundary condition is the pressure outlet, the boundary condition is the standard wall functions, and the liquid in the computational domain is seawater with a density and kinetic viscosity of 1027 kg/m$^3$ and 0.00161 kg/m·s [18]. The standard $k$-$\varepsilon$ model, standard k-$\omega$ model, and realizable $k$-$\varepsilon$ model are used to simulate the flow field of the HSMPS sampling process. The experimental results show that the calculation results of the three models are similar, within 10%. The $k$-$\varepsilon$ model is the most widely used model in engineering, and the realizable $k$-$\varepsilon$ model takes into account the rotation and curvature, so the realizable $k$-$\varepsilon$ model was finally selected as the turbulent simulation model in this study.

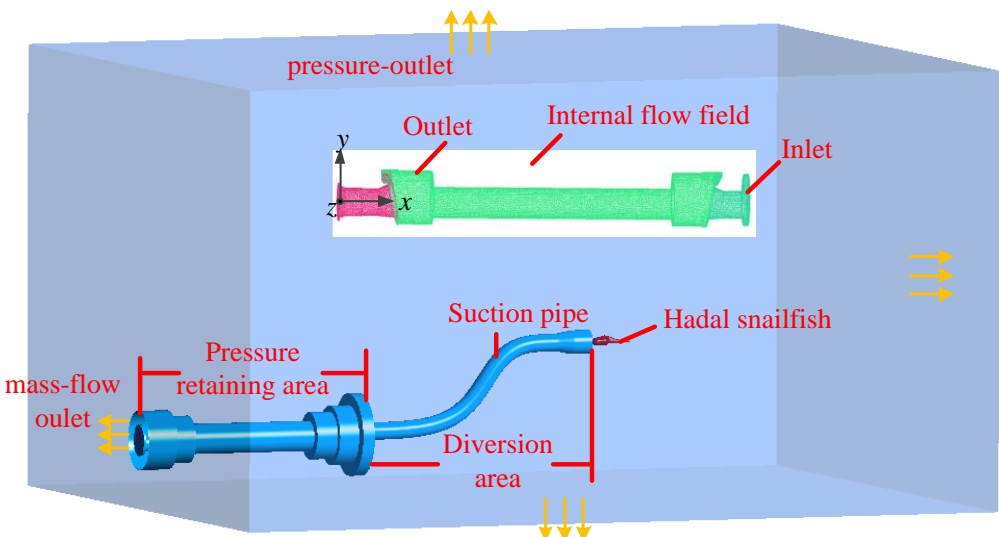

**Figure 4.** Fluid calculation model of HSMPS.

To study grid independence, three simulations with different grid numbers are compared. The suction flow rate of the HSMPS is set to 18 m$^3$/h, and the convergence rate $R_g$ can be used to verify the mesh convergence [19]. $R_g$ is given by the following formula:

$$R_g = \frac{\varepsilon_1 - \varepsilon_2}{\varepsilon_2 - \varepsilon_3} \tag{1}$$

where $\varepsilon_1$, $\varepsilon_2$, and $\varepsilon_3$ are the results of fine, medium, and coarse grids.

Table 2 shows three simulation results with different grid numbers. All the $R_g$ are in the range of 0–1, which indicates that the convergence is monotonic as the number of grids increases. To balance the simulation accuracy and time, an intermediate grid is adopted; that is, the number of grids is 5.12 million. As shown in Figure 5, the flow field around the hadal snailfish was finely drawn using a tetrahedral unstructured grid to divide the computational domain. The first grid thickness of the inner wall of the suction pipe in the diversion area is 1 mm, and the grid growth rate is 1.2. Using Fluent to improve the grid quality, 5% of the grids with quality less than 0.5 are increased to more than 0.7, and the final number of grids is 5.52 million.

**Table 2.** Verification of grid independence.

| Grid | Maximum Inlet Speed | Global Maximum Speed |
|---|---|---|
| 4259452 | 2.35 | 2.32 |
| 5117974 | 2.19 | 2.38 |
| 6122362 | 2.14 | 2.42 |
| $R_g$ | 0.31 | 0.67 |

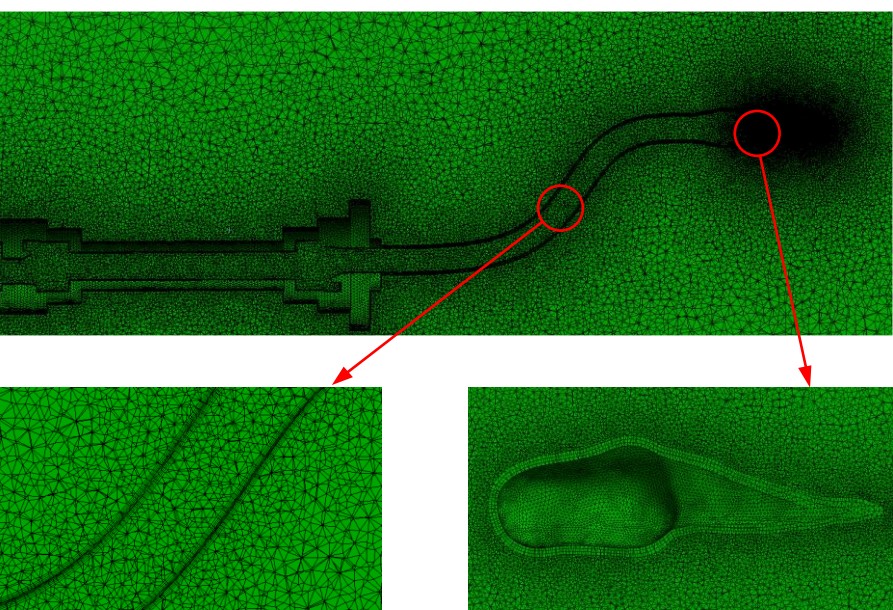

**Figure 5.** Model diagram of grid.

In the process of capturing hadal snailfish using a HSMPS on the seafloor, too large a suction speed can cause serious damage to the hadal snailfish, and with too small a suction speed, the hadal snailfish can actively swim against the current, resulting in the inability to capture the target. When the suction speed is greater than the limit flow speed of the fish, the fish cannot swim against the current, which is in the range of 0.6~1.2 m/s [20]. Therefore, this paper studies the flow field distribution of the velocity and pressure during HSMPS capture at a flow rate of 13.07 m$^3$/h (1.0 m/s) of the suction pump.

According to the law of conservation of mass, the difference between the mass of the fluid flowing into and out of the control body is equal to the amount of change in the mass of the fluid in the control body during the period, which gives the continuity equation for the fluid flow:

$$\frac{d\rho}{dt} + \frac{\partial}{\partial x_i}(\rho u_i) = 0 \tag{2}$$

According to the law of momentum, the expression for the conservation of momentum in the fluid flow processes is written in a tensor form as:

$$\frac{\partial}{\partial t}(\rho u_i) + \frac{\partial}{\partial x_i}(\rho u_i u_j) = \rho f_i + \frac{\partial}{\partial x_j}\sigma_{ij} \tag{3}$$

The realizable $k$-$\varepsilon$ model transport equation is:

$$\frac{\partial}{\partial t}(\rho k) + \frac{\partial}{\partial x_i}(\rho u_i k) = \frac{\partial}{\partial x_j}\left[\left(\mu + \frac{\mu_i}{\sigma_k}\right)\frac{\partial k}{\partial x_j}\right] + G_k + G_b - \rho\varepsilon - Y_M + S_k \tag{4}$$

$$\frac{\partial}{\partial t}(\rho\varepsilon) + \frac{\partial}{\partial x_i}(\rho u_i k) = \frac{\partial}{\partial x_j}\left[\left(\mu + \frac{\mu_i}{\sigma_\varepsilon}\right)\frac{\partial k}{\partial x_j}\right] + \frac{\varepsilon}{k}(G_k C_{3\varepsilon} G_h) - \rho C_{2\varepsilon}\frac{\varepsilon^2}{k} + S_\varepsilon G_k \tag{5}$$

where $u$ is the fluid velocity; $\rho$ is the turbulent viscosity of the fluid; $t$ is the time variable; $x_i$ and $x_j$ are the flow components; $\mu$ is the molecular viscosity $C_{1\varepsilon}$ = 1.44, $C_{2\varepsilon}$ = 1.92, and $C_{3\varepsilon}$ = 0.99; the Planck numbers $\sigma_k$ and $\sigma_\varepsilon$ are 1.0 and 1.3, respectively; $u_i$, $u_j$, and $u_k$ are the velocity components of the three coordinates; $G_k$ and $G_b$ are the mean velocity gradient and the buoyancy-induced turbulence energy generation terms, respectively; and $S_k$ and $S_\varepsilon$ are the user-defined source terms [21].

### 3.2. Flow Field Distribution

The velocity distribution of the flow field at the HSMPS pumping flow rate of 13.07 m$^3$/h is shown in Figure 6. The flow velocity at the inlet of the suction pipe in the diversion area increased from 0 to 1.3 m/s, which is greater than the limiting flow velocity of the hadal snailfish. The high-speed area of the flow field is mainly concentrated in the bending position of the inner wall of the suction pipe in the diversion area and the exit position of the deep-sea macroorganism sampler, with a maximum flow velocity of 1.7 m/s. The low-speed area is mainly concentrated in the outlet and inlet flap seal valve on both sides near the wall at the location, with a speed in the range of 0.3~0.6 m/s. The flow velocity in the pressure-retaining area is more stable, and the velocity gradient in the flow direction is small, with the flow velocity in the range of 1.0~1.3 m/s. The velocity vector diagram of the flow field of the deep-sea macroorganism sampler is shown in Figure 7. The backflow phenomenon occurs near the wall on both sides of the outlet and inlet sealing valves, which is the sudden increase in diameter when the fluid enters the outlet sealing valves from the diversion area, resulting in a small velocity near the wall. The pressure distribution of the flow field at the HSMPS pumping flow rate of 13.07 m$^3$/h is shown in Figure 8. The HSMPS internal flow field formed a certain negative pressure environment, which created favorable conditions for the hadal snailfish to enter the pressure-retaining area through the diversion area. The inevitable bending of the diversion area suction tube in the process of capturing hadal snailfish causes a large local negative pressure, which increases the possibility of a collision between the HSMPS and the bending position of the inner wall of the diversion area suction tube in the process of capturing hadal snailfish. At the inlet sealing valve, due to the sudden change in the circulation area causing local pressure loss, the negative pressure near the inlet sealing valve is small, with a minimum of 599 Pa. At the pressure-retaining area, the pressure values at each position remain stable, with pressure values in the range of 1498 to 1798 Pa. At the outlet sealing valve, the negative pressure value is generated with a maximum of 3596 Pa.

During the sampling of hadal snailfish using the HSMPS on the seafloor, speed and pressure fluctuations occur at different locations of the HSMPS. From the velocity flow field distribution diagram, it can be seen that the hadal snailfish are very likely to collide with the inner wall of the suction tube and pressure-retaining area in the process of capturing the hadal snailfish using the HSMPS. Excessive collision velocity is highly likely to cause damage to the fish epidermis, tissues, and organs [22]. From the pressure-flow field distribution diagram, it can be seen that the pressure gradients at the different

locations of the HSMPS are different. Rapid pressure changes are very likely to cause fish damage [23]. As the radial, static pressure, and velocity at each position of the HSMPS are extremely difficult to measure in the test, and the data can be easily obtained using numerical calculations, the numerical calculation results are used to analyze the distribution laws of the pressure gradient and velocity at different positions of the HSMPS under different pumping flow rates (18, 16, 14, and 12 m$^3$/h).

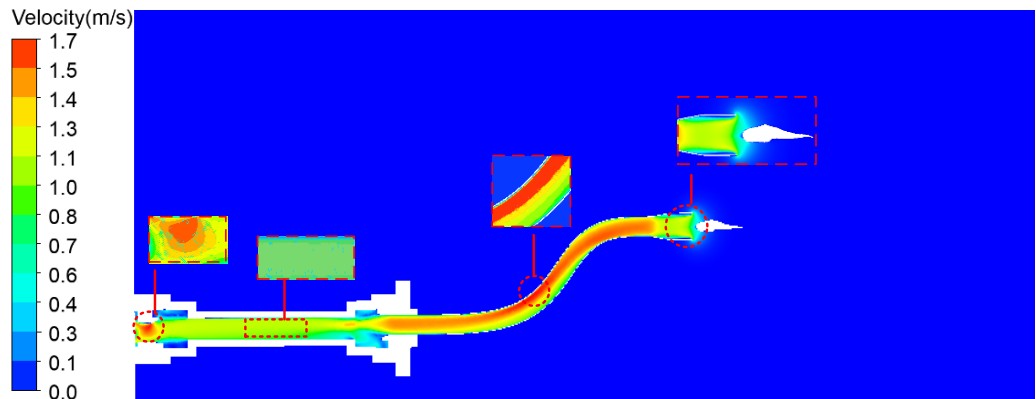

**Figure 6.** Flow field velocity distribution of HSMPS.

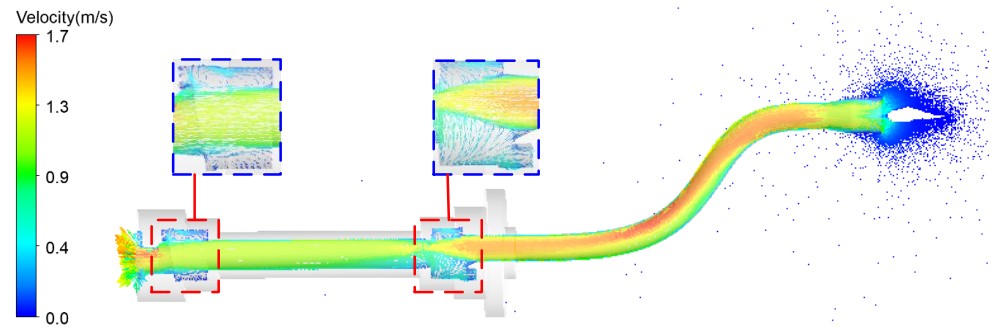

**Figure 7.** Velocity vector distribution of HSMPS flow field.

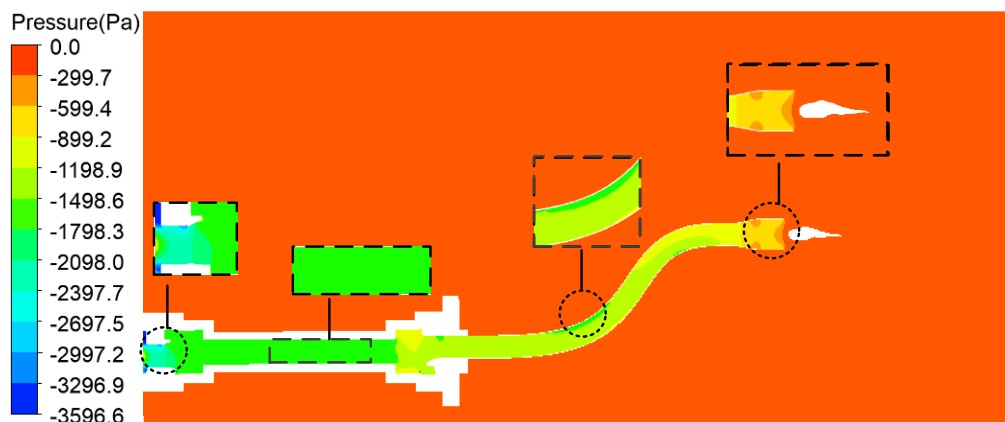

**Figure 8.** Flow field pressure distribution of HSMPS.

## 4. Results and Analysis

### 4.1. Distribution of Radial Velocity

The radial velocity distribution at each position of the HSMPS for different pumping speeds is shown in Figure 9. The greater the pumping flow rate of the HSMPS, the greater the radial velocity of each position of the HSMPS, and the maximum velocity position

appears at the end of the pumping suction area of the HSMPS. When the pumping flow rate is 18 m³/h, the maximum radial speed at the end of the pumping suction area is 2.25 m/s, and when the pumping flow rate is 12 m³/h, the maximum radial speed at the end of the pumping suction area is 1.52 m/s. The gradient of the radial velocity change in the pressure-retaining area is small because the inner diameter at each position of the pressure-retaining area is the same, and the radial velocity at each position is constant. In addition, the radial velocity of the upper wall surface is greater than the radial velocity of the lower wall surface, and the hadal snailfish are likely to collide with the upper wall surface of the pressure-retaining area during the HSMPS pumping process. For deep-sea soft gelatinous organisms, excessive collision speed is likely to cause the surface contusion of deep-sea organisms. Therefore, when designing a HSMPS, installing a layer of buffer material on the inner wall of the pressure-retaining area to reduce the possibility of deep-sea organism damage should be considered [24].

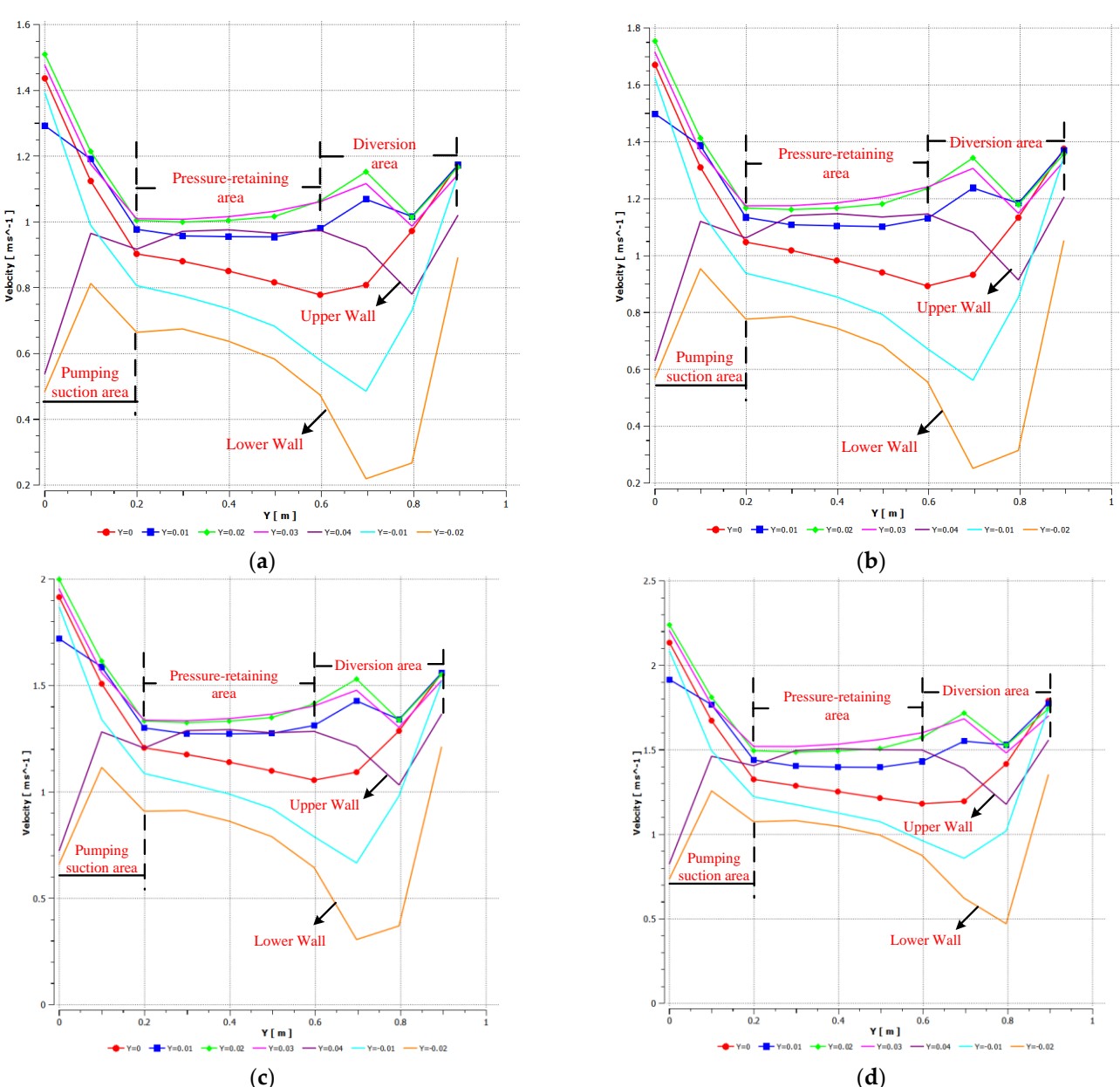

**Figure 9.** Radial velocity distribution of HSMPS. (**a**)12 m³/h, (**b**)14 m³/h, (**c**) 16 m³/h, (**d**) 18 m³/h.

When the pumping flow rate is 18 m$^3$/h, the maximum radial velocity in the pressure-retaining area is 1.52 m/s. When the pumping flow rate is 12 m$^3$/h, the maximum radial velocity at the end of the pressure-retaining area is 1.05 m/s, which is less than the limit flow rate of the fish, so the fish may swim against the current when they enter the pressure-retaining area [25]. The radial velocity variation at the lower wall of the diversion area is large, and the radial velocity variation trend at each position of the diversion area is the same. In addition, the radial velocity in the diversion area first decreases and then increases due to the eccentric design of our designed inlet seal valve, where the inner diameter of the inlet seal valve is first small and then large, and the position at the maximum inner diameter has the smallest radial velocity. At the inlet sealing valve, the radial velocity changes greatly, and shear flow easily occurs. Neitzel [26] found that at a high shear rate, it is easy to cause internal bleeding damage to the bodies of fish caused by the rupture of their swim bladders or internal organ damage. For most fish, when the shear rate is lower than 500 s$^{-1}$, the shear rate has little effect on the fish.

When the suction flow rate is 18 m$^3$/h, the minimum radial velocity in the diversion area is 0.48 m/s, and the maximum radial velocity in the diversion area is 1.75 m/s. When the pumping flow rate is 12 m$^3$/h, the minimum radial velocity in the diversion area is 0.23 m/s, and the maximum radial velocity is 1.15 m/s. The maximum radial velocity is less than the limit flow velocity of the fish, and the fish may not be captured. When the pumping flow rate is 14 m$^3$/h, the maximum radial velocity is 1.2 m/s. Therefore, we tried to pump a flow rate greater than 14 m$^3$/h in the process of pumping the hadal snailfish on the bottom to ensure the success rate of capture.

*4.2. Distribution of Radial Pressure*

The radial pressure distribution at each position of the HSMPS for different pumping speeds is shown in Figure 10. The radial pressures at each position of the HSMPS overlap, and the radial pressures in the flow field inside the HSMPS are all negative. The radial pressure first decreases and then increases as the flow field passes through the inlet sealing valve in the diversion area. Similar to the trend of radial velocity described above, the minimum radial pressure in the diversion area occurs at the location with the largest inlet seal valve bore, and the maximum radial pressure occurs at the inlet flap seal valve inlet. When the suction flow rate is 18 m$^3$/h, the minimum radial pressure in the diversion area is −2400 Pa, and the maximum radial pressure in the diversion area is −2900 Pa. When the suction flow rate is 12 m$^3$/h, the minimum radial pressure in the diversion area is −1080 Pa, and the maximum radial pressure is −1280 Pa. The Electric Power Research Institute of the United States found, through experimental research, that the sharp pressure drop on the surface of fish generally may lead to internal organ damage caused by the internal bleeding of fish and expansion of the swim bladder [27]. When the minimum pressure in the flow channel is more than 60% of the fish's adaptive environmental pressure, it is considered that the flow field pressure will not have a substantial impact on the fish [28]. Compared with the deep-sea ultra-high-pressure environment, the pressure change in the internal flow field of the HSMPS is very small, so it can be considered that the flow field of the hadal snailfish will not affect its tissues and organs before they are sucked in.

The structure of the pressure-retaining area is symmetrical around the central axis, which leads to the radial pressure distribution of the flow field in the pressure-retaining area also being symmetrical, and the radial pressure gradient is small. When the suction flow rate is 18 m$^3$/h, the radial pressure in the pressure-retaining area is around −2800~2900 Pa. When the suction flow rate is 12 m$^3$/h, the radial pressure in the pressure-retaining area is about −1250~1300 Pa, which creates favorable conditions for the hadal snailfish to enter the pressure-retaining cylinder in the pressure-retaining area through the suction pipe in the diversion area. When the width of the deep-sea creatures captured using the HSMPS is large, the deep-sea creatures form gap flow at the elbow position of the diversion, and the pressure-retaining barrel of the pressure-retaining area, which easily "blocks" the flow channel, reduces the cross-sectional area of the flow channel, increases the radial velocity

and pressure of the fluid, and easily damages the deep-sea creatures. Therefore, the size of the capture target should be considered when designing a HSMPS.

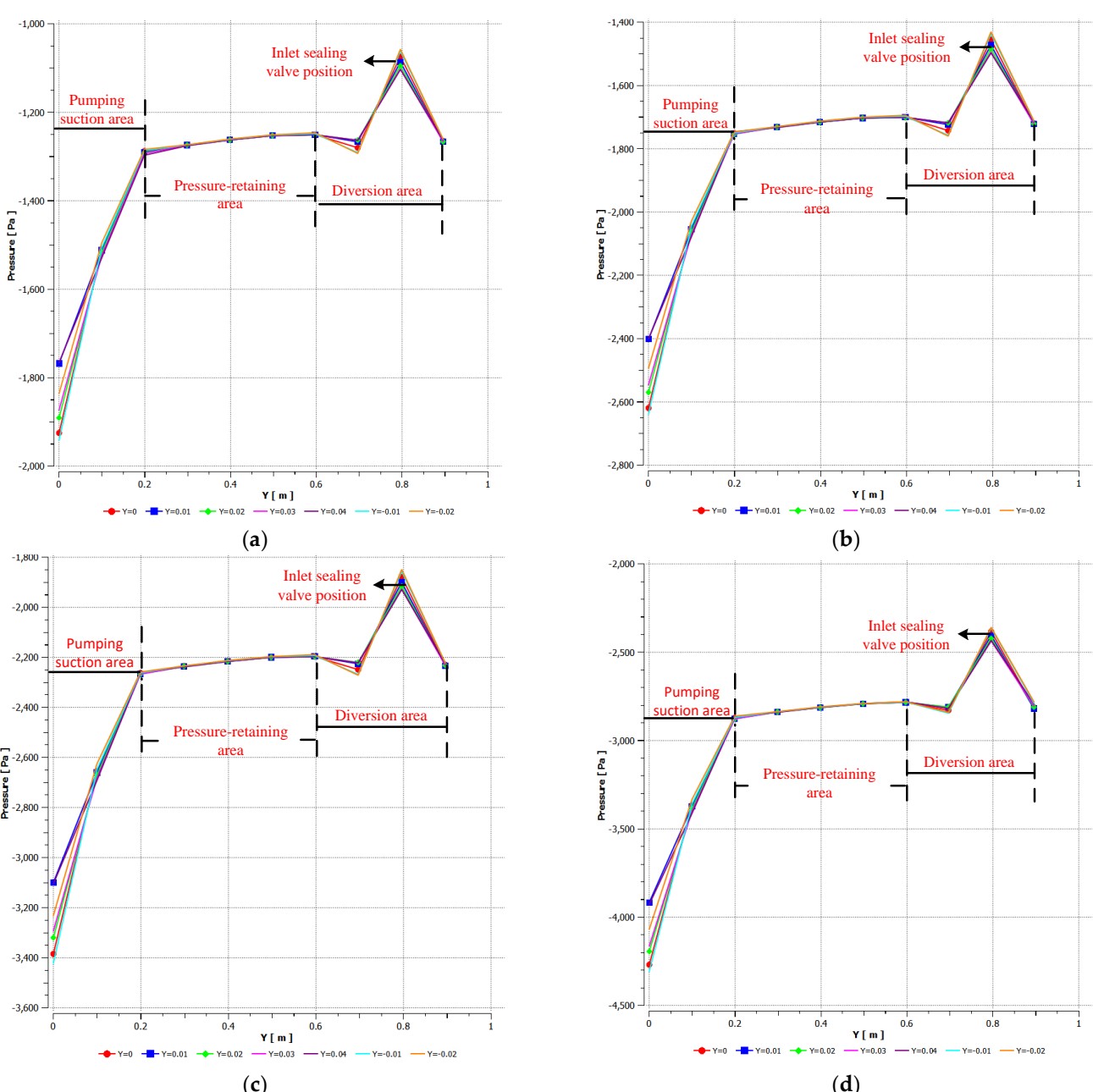

**Figure 10.** Radial pressure distribution of HSMPS. (**a**) 12 m³/h, (**b**) 14 m³/h, (**c**) 16 m³/h, (**d**) 18 m³/h.

The pressure gradient in the pumping suction area varies greatly. When the pumping flow rate is 18 m³/h, the maximum radial pressure in the pumping suction area is −4300 Pa, and when the pumping flow rate is 12 m³/h, the maximum radial pressure in the pumping suction area is −1945 Pa. As the HSMPS is equipped with a non-return device at the outlet end of the pressure-retaining area during the pumping of the hadal snailfish, the pressure change in the flow field within the pumping suction area does not affect the damage to the hadal snailfish.

### 4.3. Suction Test

To verify the feasibility of the HSMPS, a simple suction test was performed. As shown in Figure 11, the HSMPS simulation suction test steps can be divided into (1) fill the water tank with enough water, open the flap sealing valves at the inlet and outlet of HSMPS, and limit it by the trigger lever; (2) put the HSMPS and the bracket into the water tank to ensure that the water level in the water tank exceeds the height of the HSMPS, and connect the suction pump and the hydraulic pump station using hydraulic pipeline; (3) put the experimental fish into the water tank, start the hydraulic pump station, and observe whether the experimental fish can be sucked into HSMPS by controlling the different flow rates of the hydraulic pump station; and (4) turn off the hydraulic pump station, and record the survival state of the test fish under different suction flow rates. The results show that when the suction flow rate is lower than 14 m$^3$/h, the experimental fish feel the current and escape. When the suction flow is in the range of 14 m$^3$/h~16 m$^3$/h, the HSMPS can catch experimental fish, and one should check that the experimental fish can swim normally after catching. When the suction flow is in the range of 16 m$^3$/h~18 m$^3$/h, the HSMPS can catch experimental fish, but it is found that the surfaces of the experimental fish are bruised, which may be caused by the excessive suction speed and the experimental fish hitting the HSMPS. Therefore, the HSMPS should try its best to keep the pump flow between 14 and 16 m$^3$/h in the process of sucking organisms from the seabed to balance the damage and the escape speed of the deep-sea organisms.

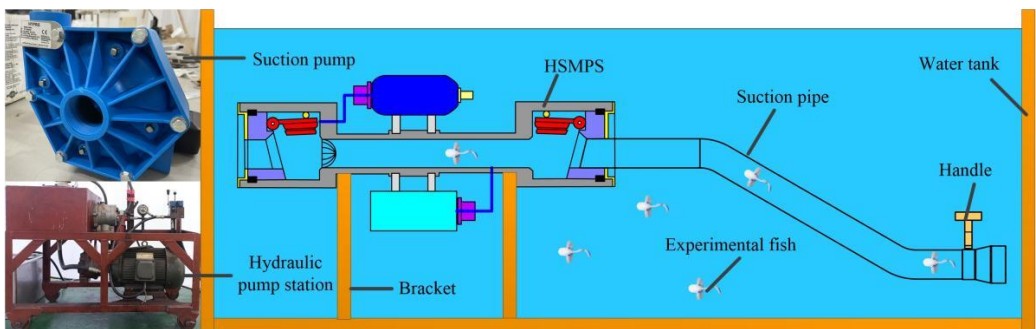

**Figure 11.** Principle diagram of suction test.

### 4.4. High-Pressure Chamber Test

To check the performance of the key components of the HSMPS in a high-pressure environment, a 110 MPa high-pressure chamber experiment was carried out on HSMPS, as shown in Figure 12. Three hydraulic cylinders are used to control the triggering action of the HSMPS, respectively, and the high-pressure cabin provides a deep-sea high-pressure environment for the HSMPS. Firstly, three hydraulic cylinders are, respectively, connected to the trigger rods of the outlet and inlet flap sealing valves and the trigger rods of the bait barrel through trigger ropes, and the hydraulic cylinders are provided with a liquid inlet and outlet. The high-pressure cabin end cover is provided with three sealing interfaces, one end of which is connected to the pressurizing system, and the other end is connected with the liquid inlet of the hydraulic cylinder. Then the high-pressure pump station is used to pressurize the high-pressure cabin. When the pressure of the high-pressure cabin reaches the test pressure, the reciprocating motion of the three hydraulic cylinders can be controlled using the pressurization system, thus completing the three triggering actions on the HSMPS. The experimental results show that the HSMPS can complete the sampling operation in the high-pressure environment of 110 MPa and keep the pressure for 3 h. The pressure in the HSMPS is 103 MPa, the pressure drops by 7 MPa, and the pressure-retaining performance is good.

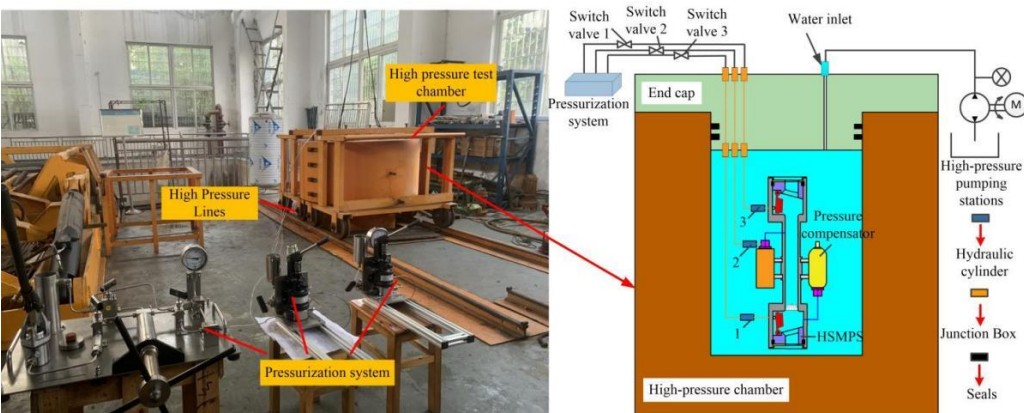

**Figure 12.** High pressure chamber test of HSMPS.

## 5. Conclusions

The structure and working principle of the HSMPS were introduced. The flow field distribution in the process of seafloor organism collection using the HSMPS was analyzed. The radial pressure and radial velocity distribution laws at different locations of the HSMPS with different pumping flows were discussed, and the following conclusions can be drawn:

(1) A full-ocean-depth hydraulic suction macrobiotic pressure-retaining sampling method is proposed, and the sampler achieves accurate sampling of seafloor organisms using pumping. The HSMPS integrates a pressure compensation mechanism, bait replenishment mechanism, and sample transfer mechanism, which can realize the pressure-retaining sampling of microorganisms at full ocean depth and can complete the sample transfer in the laboratory. The HSMPS can realize a pressure-retaining seal with one trigger of a robot, a simple structure, and a reliable seal.

(2) In the process of collecting seafloor organisms using the HSMPS, the high-speed area of the flow field is mainly concentrated in the bending position of the inner wall of the suction tube in the diversion area and the position of the outlet sealing valve in the pressure-retaining area, and the organisms easily collide with the bending position of the inner wall of the suction tube during the collection process of the HSMPS, so excessive speed should be avoided to avoid damage to the organisms as much as possible. The low-speed region is mainly concentrated in the pressure-retaining area out- and inlet sealing valves on both sides near the wall at the location, with an easy-to-produce backflow phenomenon.

(3) The radial velocity variation in the inflow area is the largest, with a maximum radial velocity variation of 1.72 m/s. When the sampling flow rate of the sampler is greater than 14 m$^3$/h, it is necessary to ensure that the organisms can be sucked into the HSMPS. The radial pressure and velocity change gradient in the pressure-retaining area is small, and the radial velocity of the upper wall surface is large, so with the HSMPS, it is very easy for the organisms to collide with the upper wall surface of the pressure-retaining area during the pumping process. The radial pressure variation in the pumping suction area is the largest, with a maximum radial pressure variation of 2355 Pa.

(4) The HSMPS was subjected to suction tests and simulated sampling tests under a 110 MPa high-pressure environment in the laboratory, and the test results show that the HSMPS was able to capture the test fish; in addition, the all-seas deep macrobiological pump suction sampler was able to complete the sampling action under a 110 MPa high-pressure environment. The test results verify the feasibility of the HSMPS design, which will provide strong support for the deep abyssal seafloor sampling operations of the full-ocean-depth manned submersible.

**Author Contributions:** Y.J. was in charge of the whole trial; G.L. wrote the manuscript; Y.P. and D.L. assisted in data collection. B.W. provided ideas on manuscript writing. All authors have read and agreed to the published version of the manuscript.

**Funding:** This work is supported by the National Key Research and Development Program of China (Grant No. 2022YFC2805904), the National Natural Science Foundation of China (Grant No. 52275106), the Postgraduate Scientific Research Innovation Project of Hunan Province (Grant No. CX20210985), and the special project for the construction of innovative provinces in Hunan (Grant No. 2020GK1021).

**Institutional Review Board Statement:** Not applicable.

**Informed Consent Statement:** Not applicable.

**Data Availability Statement:** The study did not report any data.

**Acknowledgments:** All authors thank the anonymous reviewers for constructive comments that helped improve this manuscript.

**Conflicts of Interest:** The authors declare no conflict of interest.

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
