# Peer review of "Design of a Full-Ocean-Depth Macroorganism Pressure-Retaining Sampler and Fluid Simulation of the Sampling Process"

_jmse, doi:10.3390/jmse10122007_

Round 1

Reviewer 1 Report

In this paper, the author presents a full-ocean-depth hydraulic suction macro-organisms pressure-retaining sampling method, which actively samples seafloor macro-organisms by pumping and stabilizes the pressure inside the sampler by using a pressure compensator. The flow field of the HSMPS sampling process was analyzed and the velocity and pressure distribution of the flow field at different locations of the HSMPS were obtained. The engineering prototype was used to verify the feasibility of the HSMPS design. The overall structure of the paper is rigorous, the text flow field, and the hierarchy is clear, which has important reference value for research in this field, and is innovative, meeting the technical requirements of this journal. Before acceptance, the following suggestions are made for this paper:

1. In the introduction, the author gives examples of many similar researches at home and abroad. The author can try to analyze the researches at home and abroad, enumerate their advantages and disadvantages, and compare with your own sampler to point out the advantages of it.

2. In parts 4.3 and 4.4, the author's description of the experimental process is too concise, so it is possible to try to refine the experimental steps, and it is better to analyze the experimental results, so as to make the conclusion more convincing.

Reviewer 2 Report

The reviewed manuscript concerns the design of a full-ocean-depth macro-organisms pressure-retaining sampler and fluid simulation of the sampling process. The article presents the structure and principle of operation of HSMPS. The authors analyzed the flow field distribution in the collection of seafloor organism by HSMPS. The distribution laws of radial pressure and radial velocity at different locations at different pumping flows are discussed. Before approving the article, it is worth making some additions and clarifications.
- Please specify the research objective in more detail based on the research gap.
- Please justify your choice of turbulence model.
- Has a sensitivity analysis of the numerical grid been carried out?
- The article should be supplemented with more elements of the discussion (most important note)
- Editing errors appear in several places.
